# Biosynthesis of Polyketides in *Streptomyces*

**DOI:** 10.3390/microorganisms7050124

**Published:** 2019-05-06

**Authors:** Chandra Risdian, Tjandrawati Mozef, Joachim Wink

**Affiliations:** 1Microbial Strain Collection (MISG), Helmholtz Centre for Infection Research (HZI), 38124 Braunschweig, Germany; 2Research Unit for Clean Technology, Indonesian Institute of Sciences (LIPI), Bandung 40135, Indonesia; 3Research Center for Chemistry, Indonesian Institute of Sciences (LIPI), Serpong 15314, Indonesia; tjandrawm@yahoo.com

**Keywords:** *Streptomyces*, polyketides, secondary metabolite, polyketide synthases (PKSs)

## Abstract

Polyketides are a large group of secondary metabolites that have notable variety in their structure and function. Polyketides exhibit a wide range of bioactivities such as antibacterial, antifungal, anticancer, antiviral, immune-suppressing, anti-cholesterol, and anti-inflammatory activity. Naturally, they are found in bacteria, fungi, plants, protists, insects, mollusks, and sponges. *Streptomyces* is a genus of Gram-positive bacteria that has a filamentous form like fungi. This genus is best known as one of the polyketides producers. Some examples of polyketides produced by *Streptomyces* are rapamycin, oleandomycin, actinorhodin, daunorubicin, and caprazamycin. Biosynthesis of polyketides involves a group of enzyme activities called polyketide synthases (PKSs). There are three types of PKSs (type I, type II, and type III) in *Streptomyces* responsible for producing polyketides. This paper focuses on the biosynthesis of polyketides in *Streptomyces* with three structurally-different types of PKSs.

## 1. Introduction

Polyketides, a large group of secondary metabolites, are known to possess remarkable variety, not only in their structure, and but also in their function [1,2]. Polyketides exhibit a wide range of bioactivities such as antibacterial (e.g., tetracycline), antifungal (e.g., amphotericin B), anticancer (e.g., doxorubicin), antiviral (e.g., balticolid), immune-suppressing (e.g., rapamycin), anti-cholesterol (e.g., lovastatin), and anti-inflammatory activity (e.g., flavonoids) [3,4,5,6,7,8,9]. Some organisms can produce polyketides such as bacteria (e.g., tetracycline from *Streptomyces aureofaciens*) [10], fungi (e.g., lovastatin from *Phomopsis vexans*) [11], plants (e.g., emodin from *Rheum palmatum*) [12], protists (e.g., maitotoxin-1 from *Gambierdiscus australes*) [13], insects (e.g., stegobinone from *Stegobium paniceum*) [14], and mollusks (e.g., elysione from *Elysia viridis*) [15]. These organisms could use the polyketides they produce as protective compounds and for pheromonal communication in the case for insects. 

Since the beginning of the 1940s, the history of antibiotics has greatly related to microorganisms. One of the groups of bacteria that produce many important antibiotics is Actinobacteria. Actinobacteria are Gram-positive, have high GC content, and comprise various genera known for their secondary metabolite production, such as *Streptomyces*, *Micromonospora*, *Kitasatospora*, *Nocardiopsis*, *Pseudonocardia*, *Nocardia*, *Actinoplanes*, *Saccharopolyspora*, and *Amycolatopsis* [16,17]. Their most important genus is *Streptomyces*, which has a filamentous form like fungi and has become a source of around two-thirds of all known natural antibiotics [18]. Among the antibiotics produced by *Streptomyces*, polyketides are one group of the very important compounds. Some examples of polyketides produced by *Streptomyces* are rapamycin (produced by *Streptomyces hygroscopicus*), oleandomycin (produced by *Streptomyces antibioticus*), actinorhodin (produced by *Streptomyces coelicolor*A3(2)), daunorubicin (produced by *Streptomyces peucetius*) and caprazamycin (produced *by Streptomyces* sp. MK730-62F2) [19,20,21,22,23].

Biosynthesis of polyketides is very complex because the process involves multifunctional enzymes called polyketide synthases (PKSs). The mechanism of PKS is similar to fatty acid synthase (FAS). The process includes many enzymatic reactions with different enzymes such as acyltransferase (AT), which has a role in catalyzing the attachment of the substrate (e.g., acetyl or malonyl) to the acyl carrier protein (ACP), and ketosynthase (KS), which catalyzes the condensation of substrates attached in ACP. After condensation of the substrates, the reaction continues by incorporating ketoreductase (KR), which reduces keto ester, dehydratase (DH), which dehydrates the compound, and enoylreductase (ER), which reduces the carbon-carbon double bond in the molecule (Figure 1). Unlike in FAS, the process catalyzed by KR, DH, and ER is optional in PKSs, which can give the various structures of polyketides with keto groups, hydroxy groups, and/or double bonds in different locations of the molecule [24,25,26]. In *Streptomyces*, there are three types of PKSs (type I, type II, and type III) [27,28,29]. This review describes the biosynthesis of polyketides in *Streptomyces* with three distinct types of PKSs. The focus is only on the *Streptomyces* genus because it is one of the most important producers of bioactive compounds and one of the most well-studied microbes in terms of polyketide biosynthesis. To the best of our knowledge, this is the first review that describes the three types of PKSs that are involved in the biosynthesis of polyketides in *Streptomyces*.

## 2. Polyketide Synthases Type I

The type I polyketide synthases (type I PKSs) involve huge multifunctional proteins that have many modules containing some domains, in which a particular enzymatic reaction occurs (Figure 2). Each module has the responsibility of performing one condensation cycle in a non-iterative way. Because this system works with some modules, it is also called modular PKS. The essential domains existing in each module are acyltransferase (AT), keto synthase (KS), and acyl carrier protein (ACP) which collaborate to produce β-keto ester intermediate. In addition, the other domains that may be present in the module are β-ketoreductase (KR), dehydratase (DH), and enoyl reductase (ER), which are responsible for keto group modification. In the process of producing polyketide, the expanding polyketide chain is transferred from one module to another module until the completed molecule is liberated from the last module by a special enzyme [2,26,31].

Furthermore, type I PKSs are generally responsible for producing macrocyclic polyketides (macrolides), although there was also a study reporting that type I PKSs are also involved in the biosynthesis of linear polyketide tautomycetin [32]. Macrolide belongs to a polyketide compound characterized by a macrocyclic lactone ring, which has various bioactivities such as antibacterial, antifungal, immunosuppressing, and anticancer. As an antibacterial agent, macrolide works by inhibiting protein synthesis by binding to the 50S ribosomal subunit and blocking the translocation steps of protein synthesis [8,27,33]. Some examples of macrolides produced by *Streptomyces* are rapamycin, FK506, spiramycin, avermectin, methymycin, narbomycin, and pikromycin, as shown in Figure 3 [34,35,36,37]. These compounds were produced by multifunctional polypeptides encoded by a biosynthetic gene cluster. The list of some polyketides produced by *Streptomyces* with their huge multifunctional proteins can be seen in Table 1.

### 2.1. Biosynthesis of Rapamycin

Rapamycin is a 31-membered ring macrolide produced by *Streptomyces hygroscopicus* isolated firstly from the soil of Easter Island (Chile) in the South Pacific Ocean. It is a hydrophobic compound and known as an antifungal compound against *Candida albicans*, *Cryptococcus neoformans*, *Aspergillus fumigatus*, *Fusarium oxysporum*, and some pathogenic species from the genus *Penicillium*. The antifungal mechanism of this compound has been described by diffusing into the cell and binding to intracellular receptor immunophilin FKB12. The FKBP12-rapamycin complexes inhibit enzymes required for signal transduction and cell growth. These enzymes are TOR (target of rapamycin) kinases that are conserved and very important for cell cycle progression. Interestingly, it was also reported that rapamycin has not only antifungal activity, but also anticancer and immunosuppressant activity [8,27,38,39].

Rapamycin is synthesized by type I PKS rapamycin synthase (RAPS) [40]. The rapamycin-PKS gene cluster (*rapPKS*) is 107.3 kb in size and has three remarkably large ORFs (open reading frames), *rapA*, *rapB*, and *rapC* which encode multifunctional protein RAPS1 (~900 kDa), RAPS2 (~1.07 MDa), and RAPS3 (~660 kDa), respectively. Protein RAPS1 comprises four modules for polyketide chain extension; protein RAPS2 contains six modules responsible for continuing the process of polyketide chain elongation until C-16; and RAPS3 possesses four modules that have a role in completing the polyketide fraction of the rapamycin molecule. Overall, these three giant proteins encompass 70 domains or enzymatic functions, and because of this, rapamycin PKSs are considered as the most complex multienzyme system discovered so far [26,27,34].

In rapamycin PKSs, there is a loading domain (LD) before module 1. In LD, there are three domains, i.e., coenzyme A ligase (CL), enoylreductase (ER), and acyl carrier protein (ACP) domain, which are considered to play a role in activating, reducing a free shikimic-acid-derived moiety starter unit, and finally passing it to the ketosynthase (KS) domain of the first module, respectively. The extender units required for producing rapamycin are malonyl-CoA and methylmalonyl-CoA. The mechanism of transferring from the last domain in rapamycin PKSs and cyclisation of polyketide molecule is assisted by pipecolate-incorporating enzyme (PIE), as depicted in Figure 4. This enzyme (170 kDa) is encoded by gene *rapP*, which is also located in the *rapPKS* gene cluster [26,27,34].

### 2.2. Biosynthesis of Avermectin

Avermectin is a 16-membered ring macrolide and one of the notable anthelmintic compounds produced by *Streptomyces avermitilis* [41,49]. The biosynthesis of avermectin involves type I PKSs (AVES1, AVES2, AVES3, and AVES4). AVES1 (414 kDa) contains one loading domain and two modules; AVES2 (666 kDa) consists of four modules; AVES 3 (575 kDa) comprises three modules; and AVES4 (510 kDa) has three modules. The process of avermectin biosynthesis includes assembling of the polyketide-derived initial aglycon (6, 8a-seco-6, 8a-deoxy-5-oxoavermectin aglycons) by AVES1–4, alteration of the initial aglycon to avermectin aglycons, and, as the last step, the glycosylation of avermectin aglycons to produce avermectins. The starter unit for avermectin biosynthesis is isobutyryl-CoA (derived from valine) or 2-methylbutyryl-CoA (derived from isoleucine), whereas the extender units involved in the production of avermectin are seven malonyl-CoAs (for acetate units) and five methylmalonyl-CoAs (for propionate units). The nucleotide sequence of the avermectin biosynthetic gene cluster comprises 18 ORFs spanning a distance of 82 kb, in which four large ORFs encode the avermectin polyketide synthase (AVES1, AVES2, AVES3, and AVES4) and some of the 14 ORFs encode polypeptides having important roles in avermectin biosynthesis [41].

### 2.3. Biosynthesis of Candicidin

Candicidin is a 38-membered ring polyene macrolide produced by *Streptomyces griseus* IMRU 3570 that has antifungal activity. Like the other polyene compounds, the antifungal mechanism of candicidin is also by disrupting the fungal cell membrane. Candicidin has both the amino sugar mycosamine and the aromatic component p-aminoacetophenone in its macrolide structure [43,50].

The candicidin biosynthetic gene cluster (<205 kb) was cloned and partially sequenced. Four genes, canP1, canP3, canP2 (incomplete), and canPF (incomplete), were determined as genes encoding parts of type I PKSs (CanP1, CanP2, CanP3, and CanPF). CanP1 contains one loading domain and one module; CanP2 consists of three modules; and CanP2 comprises six modules. CanPF hypothetically serves as one end of the PKS gene cluster. The starter unit is PABA (p-aminobenzoic acid), and the extender units are four methylmalonyl-CoAs and 17 malonyl-CoAs. At the end of the process in PKS, the molecule is released by thioesterase (CanT). In the next step, the compound is cyclized to become candicidin aglycone, oxidized by P450 monooxygenase (CanC) with aid from ferredoxin (CanF). The last step is glycosylation by adding mycosamine to the structure [43,44].

### 2.4. Biosynthesis of Tautomycetin

Tautomycetin, firstly isolated from *Streptomyces griseochromogenes* and then from *Streptomyces* sp. CK4412, is an antifungal compound and an activated T cell-specific immunosuppressive compound. The inhibition of T-cells’ proliferation is by the apoptosis mechanism. Unlike the other type I polyketide-derived compounds, tautomycetin has a linear structure [32,51].

The tautomycetin (TMC) biosynthetic gene cluster (~70 kb) has two ORFs that encode type I PKSs (Tmc A and TmcB). TmcA has six modules including the loading module, and TmcB has four modules, the TE (thioesterase) domain of which is located in TmcB. TE domain is responsible for releasing the intermediate chain of the compound from the PKS. The biosynthesis of TMC requires malonyl-CoA as a starter unit and the extender units such as 4 malonyl-CoAs, 4 methylmalonyl-CoAs, and 1 ethylmalonyl-CoA. After being released from PKS, the intermediate compound is modified by post-PKS mechanisms such as hydroxylation, decarboxylation, dehydration, and esterification with the cyclic C8 dialkylmaleic anhydride moiety [32].

## 3. Polyketide Synthases Type II

The type II polyketide synthases (type II PKSs) are responsible for producing aromatic polyketide. Based on the polyphenolic ring system and their biosynthetic pathways, the aromatic polyketides produced by type II PKSs generally are classified into seven groups, i.e., anthracyclines, angucyclines, aureolic acids, tetracyclines, tetracenomycins, pradimicin-type polyphenols, and benzoisochromanequinones [52]. 

Anthracyclines consists of a linear tetracyclic ring system with quinone–hydroquinone groups in rings B and C. Angucyclines have an angular tetracyclic ring system. The aureolic acids have a tricyclic chromophore. Tetracyclines contain a linear tetracyclic ring system without quinone–hydroquinone groups in rings B and C. Tetracenomycins have a linear tetracyclic ring system with the quinone group in ring B. Pradimicin-type polyphenols are considered as extended angucyclines. Benzoisochromanequinones contain a quinone derivative from the isochroman structure [52]. Some examples of aromatic polyketide produced by *Streptomyces* are actinorhodin (benzoisochromanequinones), doxorubicin (anthracyclines), jadomycin B (angucyclines), oxytetracycline (tetracyclines), mithramycin (aureolic acids), tetracenomycin C (tetracenomycins), and benastatin A (pradimicin-type polyphenols) (Figure 5) [28,52,53,54,55,56,57].

Unlike type I PKSs that involve huge multifunctional proteins that have many modules containing domains and perform the enzymatic reaction in a non-iterative way, the type II PKSs have monofunctional polypeptides and work iteratively to produce aromatic polyketide. However, like the type I PKS, the type II PKSs also comprise the acyl carrier protein (ACP) that functions as an anchor for the nascent polyketide chain. In addition to possessing ACP, the type II PKSs also consists of two ketosynthases units (KS_α_ and KS_β_) that work cooperatively to produce the poly-β-keto chain. The KS_α_ unit catalyzes the condensation of the precursors; on the other hand, the role of KS_β_ in the type II PKSs is as a chain length-determining factor. The three major systems (ACP, KS_α_, and KS_β_) are called “minimal PKS” that work iteratively to produce aromatic polyketide. The other additional enzymes such as ketoreductases, cyclases, and aromatases cooperate to transform the poly-β-keto chain into the aromatic compound core. Furthermore, the post-tailoring process is conducted by oxygenases and glycosyl and methyl transferases [52,58,59,60]. The list of some aromatic polyketides produced by *Streptomyces* with their type II PKSs can be seen in Table 2.

### 3.1. Biosynthesis of Doxorubicin

Doxorubicin was isolated from *Streptomyces peucetius* in the early of 1960s. It belongs to anthracyclines that have a tetracyclic ring containing quinone and a hydroquinone group in their structure. Doxorubicin is one of the important drugs for the treatment of cancer such as breast cancer, childhood solid tumors, soft tissue sarcomas, and aggressive lymphomas. There are some proposed mechanisms for how doxorubicin kills the cancer cells: (i) intercalation of DNA and interference of topoisomerase-II-mediated DNA repair; and (ii) formation of free radicals and their deterioration of cell components such as cellular membranes, DNA, and proteins [61,62,63].

Daunorubicin (DNR)-doxorubicin (DXR) type II PKSs, encoded by *dps* genes in *Streptomyces peucetius*, are involved in the formation of doxorubicin. The biosynthesis of doxorubicin requires one propionyl-CoA as the starter unit and nine malonyl-CoAs as the extender units. The process involves two “minimal PKSs” (DpsC-DpsD-DpsG and DpsA-DpsB-DpsG) to produce a 21-carbon decaketide as an intermediate compound. The repetitive process is conducted by KS_α_ (DpsA), KS_β_ (DpsB), and ACP (DpsG). The next process employs several enzymes such as ketoreductase (DpsE), cyclases (DpsF, DpsY, and DnrD), oxygenase (DnrG and DnrF), and methyl transferase (DnrC) to produce ε-rhodomycinone, an important intermediate of doxorubicin biosynthesis. The remaining steps to synthesize doxorubicin utilize glycosyltransferase (DnrS) with the thymidine-diphospho (TDP) derivative of l-daunosamine, methyl esterase (DnrP), oxygenase (DoxA), and methyl transferase (DnrK) (Figure 6) [60,64,65,66,67,68].

### 3.2. Biosynthesis of Medermycin

Medermycin is a benzoisochromanequinone (BIQ) antibiotic, isolated from *Streptomyces* sp. K73. It has high activity against some Gram-positive bacteria such as *Staphylococcus aureus*, *Staphylococcus epidermidis*, *Sarcina lutea*, *Bacillus subtilis*, and *Bacillus cereus*. Besides antibiotic activity, medermycin also has potent activity as a platelet aggregation inhibitor. Because of its unique ability to give different colors in acidic and alkaline aqueous solution, medermycin is considered as an indicator type antibiotic [69,70,84].

Biosynthesis of medermycin requires eight malonyl-CoAs and a sugar molecule, angolosamine, which is derived from the deoxyhexose (DOH) pathway. In the first step of biosynthesis, the minimal PKS that consists of ACP (encoded by the *med*-ORF23 gene), KS_α_ (encoded by *med*-ORF1), and KS_β_ (encoded by *med*-ORF2) forms an octaketide moiety. The next process employs several enzymes such as keto reductase, aromatase, cyclase, enoyl reductase, and oxygenase/hydroxylase to produce the aglycone compound dihydrokalafungin. The aglycone structure then is combined by C-glycosyl transferase with an angolosamine structure to yield the final structure medermycin [69]. 

### 3.3. Biosynthesis of Hedamycin

Hedamycin is a pluramycin antitumor antibiotic, produced by *Streptomyces griseoruber*. This aromatic polyketide has a planar anthrapyrantrione chromophore, two amino sugars in its structure (α-l-*N,N*-dimethylvancosamine and β-d-angolosamine), and a bisepoxide-containing a side chain. The compound could inhibit 50% of human cancer cell growth at a subnanomolar concentration in three days. It is a monofunctional DNA alkylating agent, and because of its low therapeutic index, hedamycin is not clinically used [81,85].

Biosynthesis of hedamycin uses twelve malonyl-CoAs and two amino sugars, vancosamine and an angolosamine moiety. The minimal type II PKSs of hedamycin biosynthesis consist of HedC (KS_α_), HedD (CLF), and HedE (ACP). Uniquely, the initial process involves type I PKSs (HedT and HedU proteins) that produce the 2,4-hexadienyl primer unit from three malonyl-CoAs, and then, it is transferred to the minimal type II PKSs of hedamycin biosynthesis. After that, a dodecaketide structure is formed by processing nine malonyl-CoAs. The structure then is modified with keto reductase, aromatase/cyclase, and oxygenase into the aglycone compound. In the last step, two glycosyltransferases are used for incorporating two amino sugars to produce hedamycin [80,81].

### 3.4. Biosynthesis of Fredericamycin

Fredericamycin, isolated from *Streptomyces griseus* ATCC 49344, is an aromatic polyketide that contains a spirocyclic structure. It has moderate antitumor and cytotoxic activity in various cell lines. These bioactivities are suggested because of the blockage of topoisomerases I and II or the peptidyl-prolyl cis-trans isomerase Pin1 [83].

The biosynthesis of fredericamycin employs the minimal type II PKSs that contains KS_α_ (FdmF and FdmS), KS_β_ (FdmG), and ACP (FdmH). There are two alternative mechanisms for chain initiation in the biosynthesis of fredericamycin. The first one requires acetyl-CoA and two malonyl-CoAs to produce the hexadienyl-priming unit. The second mechanism is by utilizing butyryl- or crotonyl-CoA and one malonyl-CoA to yield the hexadienyl-priming unit. The next step is carried out by processing twelve malonyl-CoAs as extender units to give the pentadecaketide intermediate, and then, the cyclases and oxygenases modify the intermediate compound into the final product [82,83,86].

## 4. Polyketide Synthases Type III

Unlike the type I and type II PKSs, the type III PKSs do not utilize ACP as an anchor for the production of polyketide metabolite. In this case, acyl-CoAs are used directly as substrates for generating polyketide compounds. In order to create polyketides, this system contains enzymes that construct homodimers and catalyzes many reactions such as priming, extension, and cyclization in an iterative way. With this fact, the type III PKSs are the simplest structures among the other types of PKSs. The type III PKSs found in bacteria were reported the first time in1999, and before that time, the type III PKSs were known only to be detected in plants [87,88,89].

Some studies previously revealed that type III PKSs could also be identified in the *Streptomyces* such as RppA, found in *Streptomyces griseus*, which is responsible in the synthesis of 1,3,6,8-tetrahydroxynaphthalene (THN) [90]. Gcs, identified in *Streptomyces coelicolor* A3(2), is reported to have an important role in the biosynthesis of germicidin [91]. SrsA*,* encoded by the *srsA* gene and isolated from *Streptomyces griseus*, is known to have an important role in the biosynthesis of phenolic lipids, i.e., alkylresorcinols and alkylpyrones [29].

The type III PKS Ken2, isolated from *Streptomyces violaceoruber*, was suggested to be involved in the production of 3,5-dihydroxyphenylglycine (3,5-DHPG). This compound is a nonproteinogenic amino acid needed for the formation of kendomycin and several other glycopeptide antibiotics such as balhimycin, chloroeremomycin, and also vancomycin [92]. Cpz6, encoded by the *cpz6* gene and isolated from *Streptomyces* sp. MK730–62F2, was reported to be engaged in the biosynthesis of caprazamycins by producing a group of new triketidepyrenes (presulficidins) [93]. Moreover, another finding also suggested that DpyA catalyzes the formation of alkyldihydropyrones in *Streptomyces reveromyceticus* (Figure 7) [94].

### 4.1. Biosynthesis of Germicidin

Germicidin, a pyrone-derived polyketide, is produced by a type III PKS germicidin synthase (Gcs) and is known to inhibit spore germination. Germicidin A, produced by *Streptomyces viridochromogenes* and *Streptomyces coelicolor*, prevents the spore germination reversibly at a very low concentration (40 pg/mL). The mechanism of inhibition is suggested by affecting the sporal respiratory chain and blocking Ca^2+^-activated ATPase, thus resulting in inadequate energy for spore germination. Furthermore, germicidin A also has antibacterial properties against various Gram-positive bacteria [95,96].

Although many bacterial type III PKSs use only malonyl-CoA as both starter and extender units, the type III PKS Gcs, which is responsible for germicidin biosynthesis, is suggested to have the ability to utilize either acyl-ACP or acyl-CoA such as medium-chain acyl-CoAs (C4–C8) as starter units and malonyl-CoA, methylmalonyl-CoA, and ethylmalonyl-CoA as extender units [97,98]. In the first step, the starter unit is transacylated onto the cysteine residue of Gcs, and then, Gcs catalyzes the condensation reaction between the starter unit and extender unit concomitantly with the decarboxylation process, resulting in β-ketoacyl-CoA. The process continues with β-ketoacyl-CoA, which transacylates back onto the cysteine residue of Gcs (repetitive process) and subsequently undergoes a condensation reaction with either methylmalonyl-CoA or ethylmalonyl-CoA simultaneously with decarboxylation to formulate β,δ-diketothioester of CoA or a triketide intermediate. In the end of the reaction, cyclization of the β,δ-diketothioester of CoA is carried out to produce various types of germicidins (Figure 8) [91].

### 4.2. Biosynthesis of Tetrahydroxynaphthalene

Tetrahydroxynaphthalene or THN is a small aromatic compound that is produced by utilizing type III PKSs (RppA). The biosynthesis process of THN requires five molecules of malonyl-CoA to form a pentaketide intermediate structure, and then, it is cyclized and aromatized to yield THN product. Spontaneous oxidation of THN may result flaviolin (red pigment) [90,99,100].

### 4.3. Biosynthesis of Dihydroxyphenylglycine

In order to synthesize 3,5-dihydroxyphenylglycine (3,5-DHPG), four malonyl-CoAs are needed, and the process is catalyzed by type III PKS (Ken2 or DpgA), which leads to the formation of the intermediate tetraketide compound. The tetraketide further is modified by hydratase/dehydratase, and oxidase/thioesterase to form 3,5-dihydroxyphenylacetic acid. The final step involves transaminase and tyrosine, as the amino group donor, to yield 3,5-DHPG, which is known as a nonproteinogenic amino acid [92,100,101].

### 4.4. Biosynthesis of Alkylresorcinol

The alkylresorcinol biosynthesis in *Streptomyces griseus* is catalyzed by SrsA. The reaction needs fatty acid (starter unit), one methylmalonyl-CoA, and two malonyl-CoAs (extender unit), and the intermediate structure is tetraketide. The tetraketide structure then transforms into the aromatic compound nonenzymatically (alkylresorcinol). This reaction may occur because of the nucleophilic attack on the thioester group by the methine carbon of the intermediate tetraketide compound [29].

## 5. Conclusions

*Streptomyces* has various systems in order to produce polyketides with different structures and functions. Knowing the polyketide structures, activities, producing enzymes, starter units, extender units, and the structural genes are very important in the development of new drugs. Some mechanisms of polyketide biosynthesis in *Streptomyces* that have been reported previously could provide strong basic knowledge not only for the biosynthesis investigation of the new polyketides, but also engineering the producing system in the future.

## Figures and Tables

**Figure 1 microorganisms-07-00124-f001:**
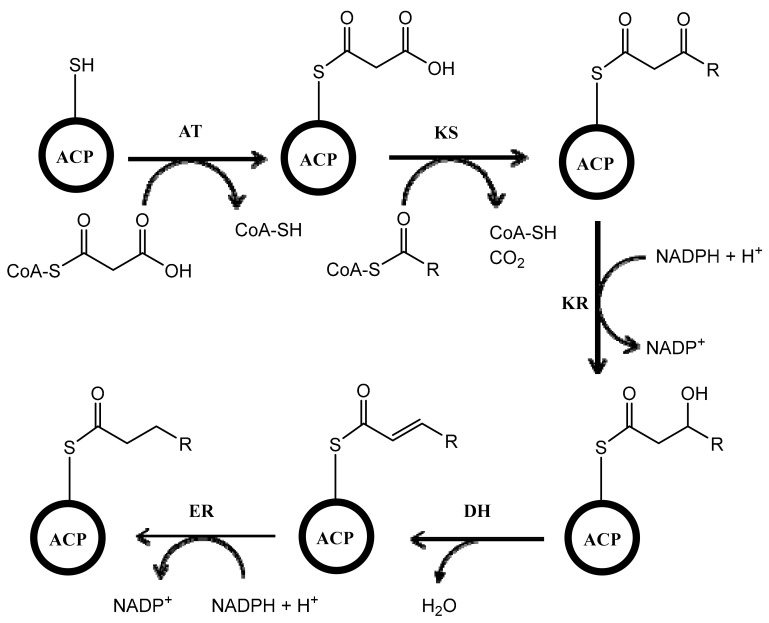
Scheme of the reaction occurring in polyketide synthases (PKSs). ACP, acyl carrier protein; AT, acyltransferase; KS, ketosynthase; KR, ketoreductase; DH, dehydratase; ER, enoylreductase. Adapted with permission of Portland Press, from Vance, S.; Tkachenko, O.; Thomas, B.; Bassuni, M.; Hong, H.; Nietlispach, D.; Broadhurst, W. Sticky swinging arm dynamics: studies of an acyl carrier protein domain from the mycolactone polyketide synthase. *Biochem. J.*
**2016**, *473*, 1097–1110 [30].

**Figure 2 microorganisms-07-00124-f002:**
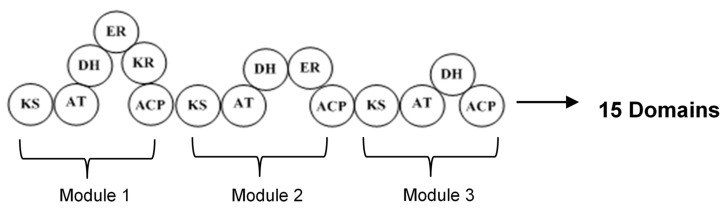
Structure of type I PKSs with three modules and 15 domains. ACP, acyl carrier protein; AT, acyltransferase; KS, ketosynthase; KR, ketoreductase; DH, dehydratase; ER, enoylreductase.

**Figure 3 microorganisms-07-00124-f003:**
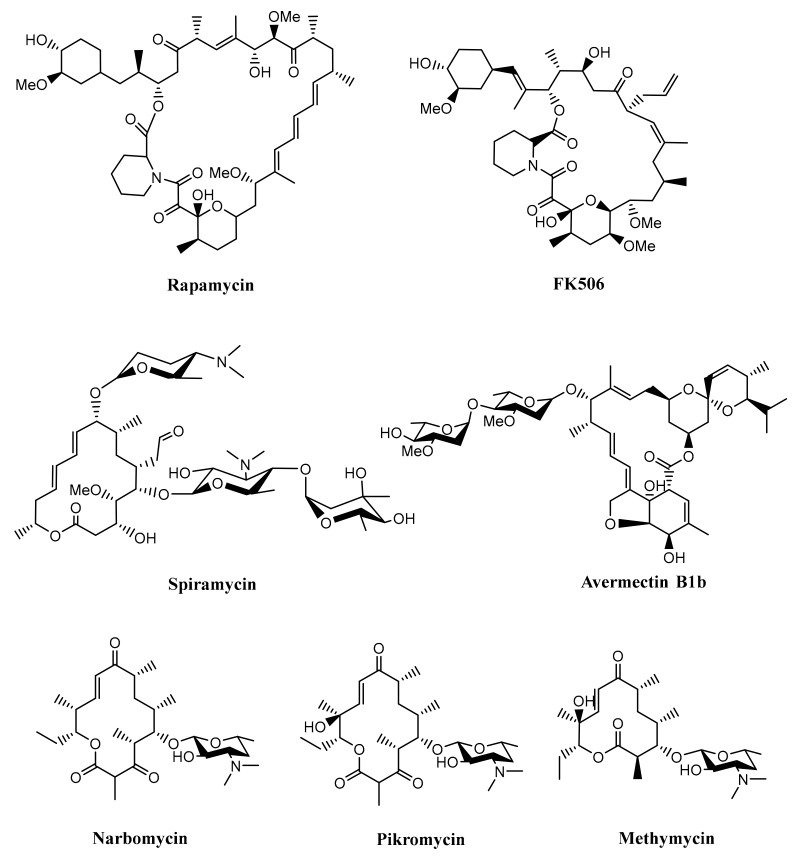
Some of the macrolides produced by *Streptomyces*.

**Figure 4 microorganisms-07-00124-f004:**
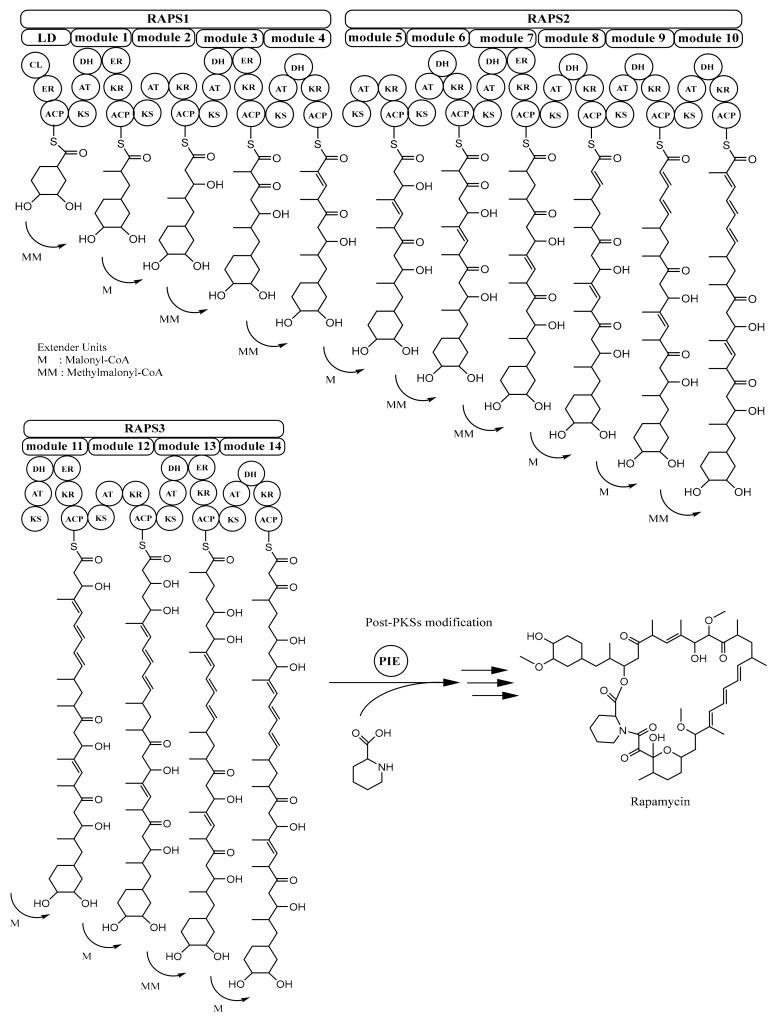
Biosynthesis of rapamycin. ACP, acyl carrier protein; AT, acyltransferase; KS, ketosynthase; KR, ketoreductase; DH, dehydratase; ER, enoylreductase; PIE, pipecolate-incorporating enzyme. Adapted with permission from Schwecke, T.; Aparicio, J.F.; Molnár, I.; König, A; Khaw, L.E.; Haydock, S.F.; Oliynyk, M.; Caffrey, P.; Cortés, J.; Lester, J.B. The biosynthetic gene cluster for the polyketide immunosuppressant rapamycin. Proc. Natl. Acad. Sci. USA 1995, 92, 7839–7843, doi:10.1073/pnas.92.17.7839 [34]. Copyright (1995) National Academy of Sciences, U.S.A. Adapted with permission of The Royal Society of Chemistry 2001, from Staunton, J.; Weissman, K.J. Polyketide biosynthesis: A millennium review. *Nat. Prod. Rep.*
**2001**, *18*, 380–416 [26]; permission conveyed through Copyright Clearance Center, Inc.

**Figure 5 microorganisms-07-00124-f005:**
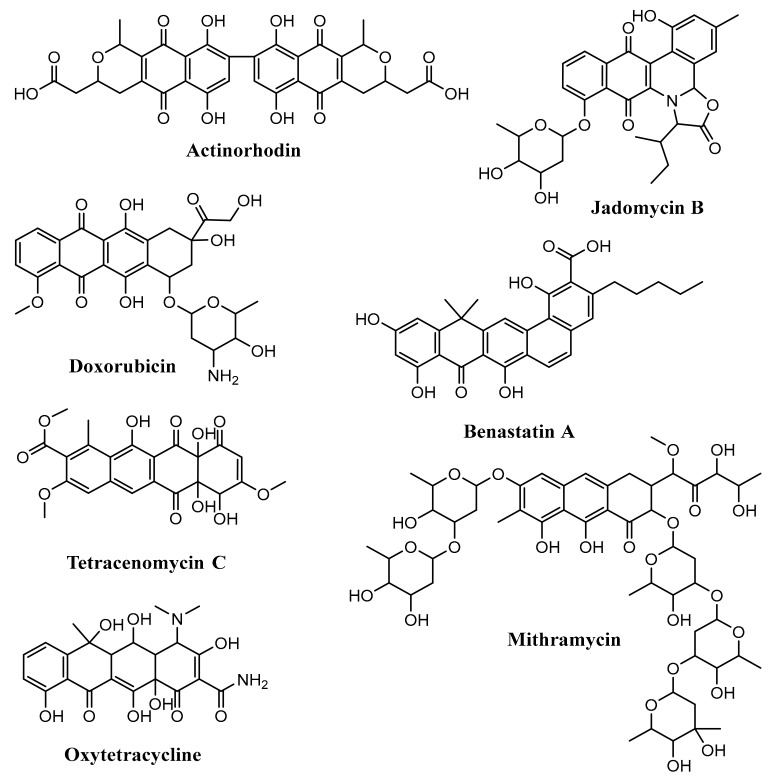
Some aromatic polyketides produced by *Streptomyces*.

**Figure 6 microorganisms-07-00124-f006:**
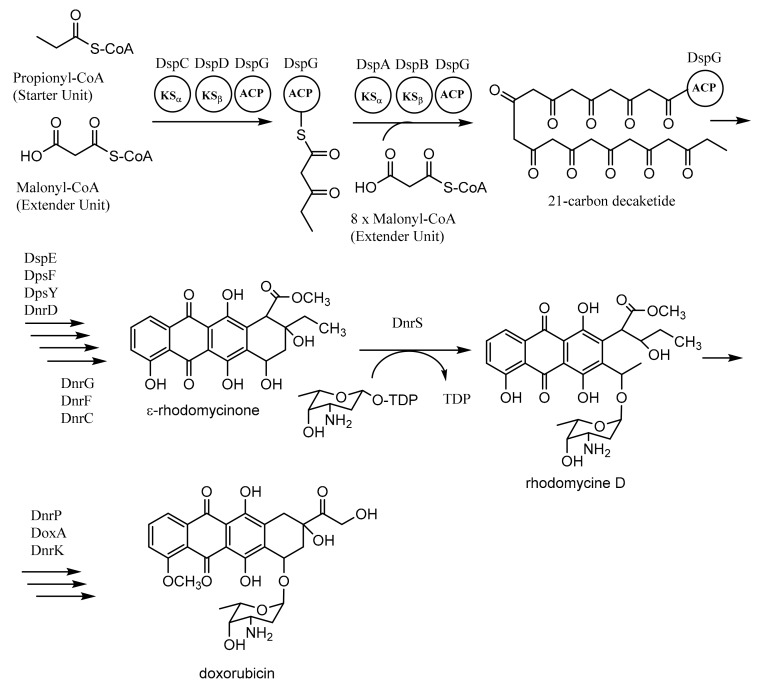
Biosynthesis of doxorubicin. Adapted with permission of The Royal Society of Chemistry 2009, from Chan, Y.A.; Podevels, A.M.; Kevany, B.M.; Thomas, M.G. Biosynthesis of polyketide synthase extender units. *Nat. Prod. Rep.*
**2009**, *26*, 90–114 [60]; permission conveyed through Copyright Clearance Center, Inc.

**Figure 7 microorganisms-07-00124-f007:**
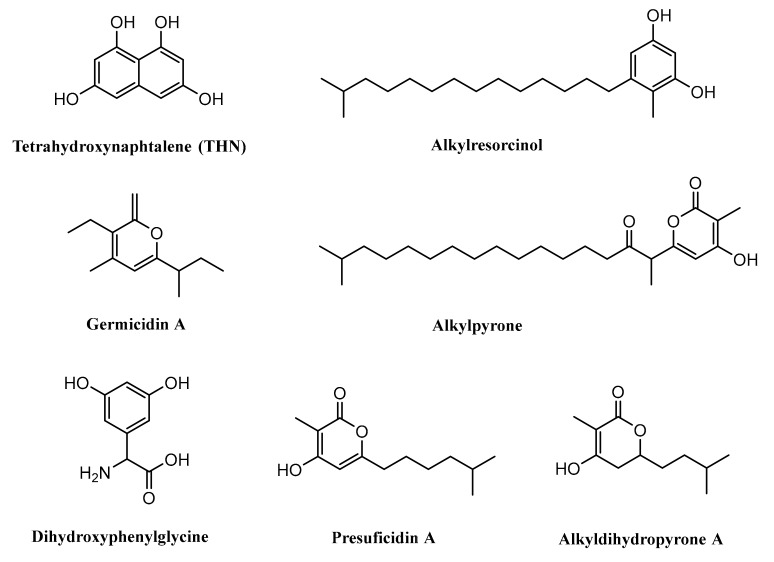
Some compounds produced by type III PKSs.

**Figure 8 microorganisms-07-00124-f008:**
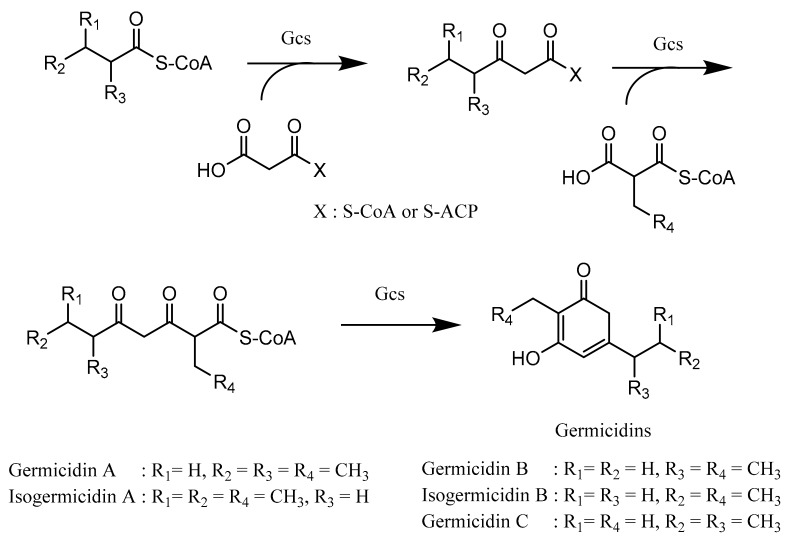
Biosynthesis of germicidins. Gcs: germicidin synthase. Adapted with permission from Song, L.; Barona-Gomez, F.; Corre, C.; Xiang, L.; Udwary, D.W.; Austin, M.B.; Noel, J.P.; Moore, B.S.; Challis, G.L. Type III polyketide synthase β-ketoacyl-ACP starter unit and ethylmalonyl-coA extender unit selectivity discovered by Streptomyces coelicolor genome mining. *J. Am. Chem. Soc.*
**2006**, *128*, 14754–14755 [91]. Copyright 2006 American Chemical Society.

**Table 1 microorganisms-07-00124-t001:** Some polyketides produced by *Streptomyces* and their type I PKSs.

Polyketide	Structure	Producer	Type I PKSs	Ref.
Avermectin	16-membered ring macrolide	*Streptomyces avermitilis*	AVES1-4	[41]
Chalcomycin	16-membered ring macrolide	*Streptomyces bikiniensis*	ChmGI-V	[42]
Candicidin	38-membered ring polyene macrolide	*Streptomyces griseus* IMRU 3570	CanP1-3, and CanPF	[43,44]
FK506 (Tacrolimus)	23-membered ring macrolide	*Streptomyces tsukubaensis, Streptomyces* sp. *MA6858*	FkbABC	[35,45]
FK520 (Ascomycin)	23-membered ring macrolide	*Streptomyces hygroscopicus* var. *ascomyceticus*	FkbABC	[46]
Methymycin, Neomethymycin, Narbomycin, Pikromycin	12-membered ring macrolide, 12-membered ring macrolide, 14-membered ring macrolide, 14- membered ring macrolide	*Streptomyces venezuelae*	PikAI-IV	[37]
Pimaricin	26-membered ring polyene macrolide	*Streptomyces natalensis*	PIMS0 and PIMS1	[47]
Rapamycin	31- membered ring macrolide	*Streptomyces hygroscopicus*	RAPS1-3	[34]
Spiramycin	16- membered ring macrolide	*Streptomyces ambofaciens*	SrmGI-V	[36]
Tautomycetin	Linear	*Streptomyces* sp. *CK4412*	TmcA and TmcB	[32]
Tylosin	16- membered ring macrolide	*Streptomyces fradiae*	TYLGI-V	[48]

**Table 2 microorganisms-07-00124-t002:** Some polyketides produced by *Streptomyces* and their minimal type II PKSs.

Polyketide	Intermediate Backbone Structure	Producer	Minimal Type II PKSs	Ref.
Medermycin	octaketide	*Streptomyces* sp. *K73*	Med-1,2,23	[69,70]
Doxorubicin	decaketide	*Streptomyces peucetius*	DpsABCDG	[60,65]
Oxytetracycline	decaketide	*Streptomyces rimosus*	OxyABCD	[71]
Gilvocarcin	decaketide	*Streptomyces griseoflavus* Gö 3592	GilABC	[72]
Oviedomycin	decaketide	*Streptomyces antibioticus*	OvmPKS	[73]
Chartreusin	decaketide	*Streptomyces chartreusis*	ChaABC	[74]
Cervimycin	decaketide	*Streptomyces tendae* HKI-179	CerABC	[75,76]
Resistomycin	decaketide	*Streptomyces resistomycificus*	RemABC	[77]
Chromomycin	decaketide	*Streptomyces griseus* subsp. *griseus*	CmmPKS	[78,79]
Hedamycin	dodecaketide	*Streptomyces griseoruber*	HedCDE	[80,81]
Fredericamycin	pentadecaketide	*Streptomyces griseus* ATCC 49344	FdmFSGH	[82,83]

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
