# Peer review of "Biosynthesis of Polyketides in Streptomyces"

_microorganisms, 2019, doi:10.3390/microorganisms7050124_

Round 1

Reviewer 1 Report

In this form the article is interesting and quite informative.

Figures have been created for this article or are they copies of existing figures ? In this case a reference should be given.

Quality of figure 2 has to be improved.

In "Streptomyces sp." sp. must not be in italics

Author Response

Points:

In this form the article is interesting and quite informative.

Figures have been created for this article or are they copies of existing figures ? In this case a reference should be given.

Quality of figure 2 has to be improved.

In "Streptomyces sp." sp. must not be in italics

Responses:

The figures in the article were actually have been modified from the existing figures. I also have already put the citation for figures even though I modified those. Is that right?

Quality of Figure 2 has been improved.

All of sp in Streptomyces sp has been changed.

Thank you.

Reviewer 2 Report

General Comments

This is a review article about the biosynthesis of polyketides in Streptomyces. It describes the three types of polyketide synthesases and gives examples of each produced by a Streptomyces species. 

The text is well written but has numerous minor grammatical errors which need to be corrected. The text appears to be thoroughly referenced, though the references were not checked by this reviewer. 

A criticism of this review is that the presentation is limited to compounds produced only by Streptomyces species, but the authors provide no explanation as to why this limitation was imposed. Also the authors do not present an exhaustive list of Streptomyces polyketides, but present an abridged list of polyketides with no explanation as to why or how the compounds they present were chosen. In choosing this approach, the authors have omitted a description of the pioneering example of Type I polyketide biosynthesis, that is, Erythromycin biosynthesis. 

Author Response

Points:

This is a review article about the biosynthesis of polyketides in Streptomyces. It describes the three types of polyketide synthesases and gives examples of each produced by a Streptomycesspecies. 

The text is well written but has numerous minor grammatical errors which need to be corrected. The text appears to be thoroughly referenced, though the references were not checked by this reviewer. 

A criticism of this review is that the presentation is limited to compounds produced only by Streptomyces species, but the authors provide no explanation as to why this limitation was imposed. Also the authors do not present an exhaustive list of Streptomyces polyketides, but present an abridged list of polyketides with no explanation as to why or how the compounds they present were chosen. In choosing this approach, the authors have omitted a description of the pioneering example of Type I polyketide biosynthesis, that is, Erythromycin biosynthesis. 

Responses:

I have fixed the grammatical errors.

Streptomyces is chosen because it is one of the most important producers of bioactive compounds and also has been well studied for its mechanism of polyketide biosynthesis. I have added this point to the article.

Not all polyketides are presented only some are selected based on the complete information of its biosynthesis and variation of structures. 

Erythromycin was omitted because it is produced from Saccharopolyspora erythraea, not Streptomyces.

Thank you.

Round 2

Reviewer 2 Report

Second Review:

This reviewer is not convinced of the need for a review article of this type in the literature. If the authors can build a better case in the introduction for why this review article fills a void in the literature then I would be more positive in my assessment of it.  

In their introduction the authors need to make a case that this type of review (Streptomyces polyketides only) has not been previously done. 

If another one has been done, then they must state how this review is a needed update. 

They must inform the reader of other closely related reviews or books in this field in their introduction. 

The authors should make the statement if it is true:  “Streptomyces are an important source of polyketide diversity, yet this is the first review in which the polyketides of Streptomyces are uniquely featured.”

The authors should also make this statement somewhere in the introduction: “Polyketide diversity far exceeds what can be described in depth in this review; and so we have presented a limited number of examples here. In Table S1 (supplemental table) we present a comprehensive list with references of all the polyketides that have been described so far to be produced by a Streptomyces species and we also list what other organisms have been reported to produce identical or closely related compounds.”  

Author Response

This review, to the best of our knowledge, is the first review that describes the three types of PKSs (polyketide synthases) that are involved in the biosynthesis of polyketides in Streptomyces. I have added this sentence in the last part of the introduction. 

The focus of this review is not talking about the diversity of polyketides but more to the description of how the polyketides are synthesized in Streptomyces with three types of polyketide synthases. With this reason, we will not put a lot of types of polyketides compounds in the paper. 

Thank you

Round 3

Reviewer 2 Report

The authors have provided adequate replies to my criticisms.